# Tetherin is an exosomal tether

**James R Edgar[1]\*, Paul T Manna[1], Shinichi Nishimura[2,3], George Banting[4], Margaret S Robinson[1]\***

[1]University of Cambridge, Cambridge Institute for Medical Research, Cambridge, United Kingdom; [2]Division of Bioinformatics and Chemical Genomics, Department of System Chemotherapy and Molecular Sciences, Graduate School of Pharmaceutical Sciences, Kyoto University, Kyoto, Japan; [3]Chemical Genomics Research Group, RIKEN Center for Sustainable Resource Science, Wako, Japan; [4]School of Biochemistry, University of Bristol, Bristol, United Kingdom

**Abstract** Exosomes are extracellular vesicles that are released when endosomes fuse with the plasma membrane. They have been implicated in various functions in both health and disease, including intercellular communication, antigen presentation, prion transmission, and tumour cell metastasis. Here we show that inactivating the vacuolar ATPase in HeLa cells causes a dramatic increase in the production of exosomes, which display endocytosed tracers, cholesterol, and CD63. The exosomes remain clustered on the cell surface, similar to retroviruses, which are attached to the plasma membrane by tetherin. To determine whether tetherin also attaches exosomes, we knocked it out and found a 4-fold reduction in plasma membrane-associated exosomes, with a concomitant increase in exosomes discharged into the medium. This phenotype could be rescued by wild-type tetherin but not tetherin lacking its GPI anchor. We propose that tetherin may play a key role in exosome fate, determining whether they participate in long-range or short-range interactions.

\*For correspondence: je333@ cam.ac.uk (JRE); msr12@cam.ac.uk (MSR)

**Competing interests:** The authors declare that no competing interests exist.

## Introduction

Exosomes are extracellular vesicles that have been implicated in a wide range of functions, including intercellular communication, tumour cell migration, RNA shuttling, and antigen presentation. By definition, exosomes are derived from multivesicular endosomes or multivesicular bodies (MVBs), which contain intralumenal vesicles (ILVs). When the MVBs fuse with the plasma membrane, the ILVs are discharged, and the resulting extracellular vesicles are called exosomes. However, there are other types of extracellular vesicles, such as those that are produced by shedding from the plasma membrane, and at present there is no standard way of specifically purifying exosomes. Thus, some of the functions that have been attributed to exosomes may need to be reassessed, because of the possibility of contamination with other types of vesicles (*Raposo and Stoorvogel, 2013*).

In a recent screen for regulators of clathrin-mediated endocytosis in HeLa cells, we observed that knocking down or inactivating the vacuolar ATPase (V-ATPase) caused the cells to produce clusters of extracellular vesicles (*Kozik et al., 2013*). These vesicles had the characteristic appearance of exosomes, suggesting that when endosomes are unable to acidify, they have an increased tendency to fuse with the plasma membrane (see Figure 6 in *Kozik et al., 2013*), a phenomenon also reported by others (*Alvarez-Erviti et al., 2011*; *Danzer et al., 2012*). Knocking down or inhibiting the V-ATPase also caused a block in clathrin-mediated endocytosis, and we proposed that this block was due to a redistribution of cholesterol from the plasma membrane to an endosomal compartment. Our hypothesis was supported by the finding that we could partially rescue the phenotype by adding exogenous cholesterol to the cells. One question raised by our study was why, if cholesterol-rich non-acidified endosomes fuse with the plasma membrane, does the plasma membrane not

**eLife digest** Cells generally communicate with each other over short distances by direct contact, and over long distances by releasing chemicals such as hormones. But there is also a third way that is less well understood – small capsules or "vesicles" called exosomes can transfer molecules from one cell to another. Exosomes are involved in the immune response and have been linked to a number of diseases, including cancer and neurodegeneration. However, scientists are still trying to understand how exosomes are made, what they contain and how they are released from cells.

A common set of cells used in laboratory studies are known as HeLa cells. These cells are the descendants of cancerous cells taken from a patient called Henrietta Lacks in 1951. When treated with a particular drug, HeLa cells produce vesicles that look like exosomes. Yet instead of moving freely like other exosomes, these structures stick together in clusters. This raises questions – are these cancer cell vesicles truly exosomes? And if so, why and how are they tethered to the cell?

Using electron microscopy and biochemical tests, Edgar et al. confirm that the unusual vesicles produced by HeLa cells are exosomes. As well as sharing characteristics with other exosomes, the vesicles also show similarities with viruses like HIV, which attach themselves to cell surfaces and each other using a protein called tetherin. Using a technique called gene editing to remove tetherin from HeLa cells allowed the exosomes in the cluster to move apart.

Further investigation revealed that some cells in the immune system also produce exosome clusters and that these clusters also contain tetherin. Edgar et al. propose that cells control whether exosomes are involved in short-range or long-range communication by controlling the amount of tetherin they produce.

So far, studies into the roles that exosomes play in the body have been hampered by a lack of experimental tools. The study by Edgar et al. opens up new methods of investigation by providing ways of altering the number of exosomes released from a cell. This should help to clarify what exosomes do and how they work in a wide range of different cell types.

regain its cholesterol? Indeed, when we measured cell surface-associated cholesterol by light microscopy, using filipin as a cholesterol probe, the loss upon V-ATPase knockdown or treatment with the V-ATPase inhibitor Bafilomycin A1 (BafA1) was only partial (~50%). In contrast, others have shown that treating cells with methyl-β-cyclodextrin, which has a similar effect on clathrin-mediated endocytosis, removes nearly all of the plasma membrane cholesterol (*Rodal et al., 1999*).

We initiated the present study to try to answer some of the questions posed by our previous study. We started by quantifying the amount of plasma membrane cholesterol more precisely by developing a method for localising cholesterol at the electron microscope level. Next, we characterised the extracellular vesicles that are produced when V-ATPase is inactivated, by labeling for exosomal markers. Finally, we investigated why the vesicles remain aggregated and associated with the plasma membrane instead of diffusing away.

## Results

### Cholesterol accumulates on extracellular vesicles following BafA1 treatment

In our previous study, we concluded that in the absence of V-ATPase activity, cholesterol accumulates in endosomal compartments, based on immunofluorescence double labelling with the cholesterol probe, filipin, and various endosomal markers (*Kozik et al., 2013*). To visualise these compartments at the ultrastructural level, we used correlative light and electron microscopy (CLEM). BafA1-treated HeLa cells (*Figure 1—figure supplement 1A*) were stained with filipin, imaged by light microscopy, and then prepared for electron microscopy. The structures that stained most intensely with filipin were found to correspond to MVBs, packed full of ILVs (*Figure 1A*).

So far, the only published studies showing cholesterol localisation at the electron microscope level have been carried out using a cleaved and a biotinylated form of the toxin perfringolysin O

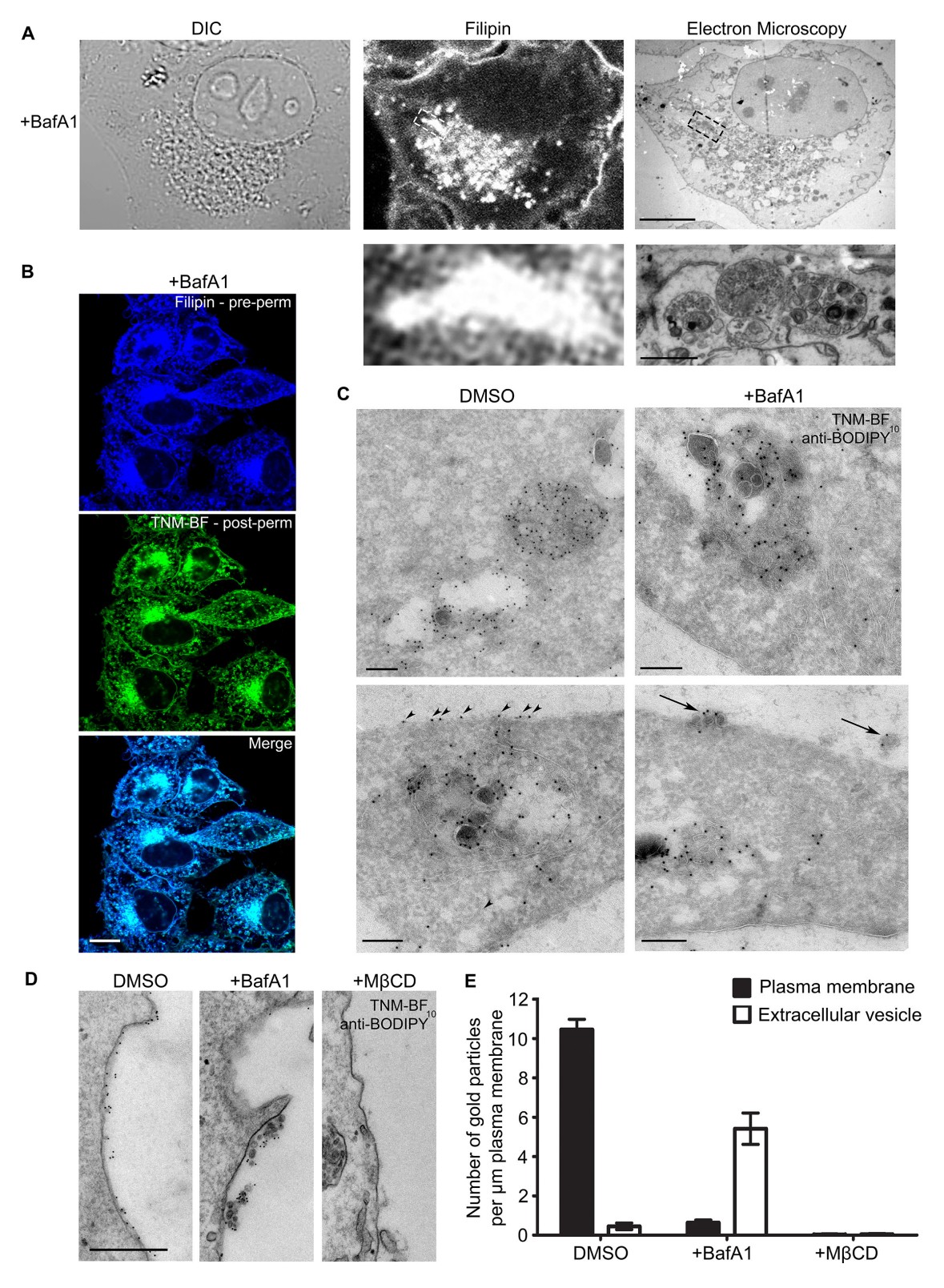

**Figure 1.** BafA1 treatment causes cholesterol to accumulate in intralumenal vesicles of multivesicular bodies and to be lost from the plasma membrane. (**A**) HeLa cells were treated with BafA1 (100 nM, 16 hr), then fixed, stained with the cholesterol probe filipin, and prepared for correlative light and electron microscopy (CLEM). Scale bars: 10 μm (upper) and 1 μm (lower). (**B**) BafA1-treated cells (100 nM, 16 hr) were stained with filipin, then permeabilised and stained with TNM-BF. Scale bar: 20 μm. (**C**) Ultrathin cryosections of mock-treated and BafA1-treated HeLa cells (100 nM, 16 hr)

*Figure 1 continued on next page*

*Figure 1 continued*

were labelled with TNM-BF and stained with rabbit anti-BODIPY followed by 10 nm protein A-gold. There is labelling both in endosomes (upper panels) and at the cell surface (lower panels). Scale bars: 200 nm. (D) Intact mock-treated, BafA1-treated (100 nM, 16 hr), or MβCD-treated (10 mM, 30 min) HeLa cells were labelled with TNM-BF followed by anti-BODIPY and protein A-gold, revealing surface cholesterol localisation. Scale bar: 500 nm. (E) Quantification of TNM/BF/anti-BODIPY gold labeling density for DMSO-treated, BafA1-treated (100 nM, 16 hr), or MβCD-treated (10 mM, 30 min) cells. Graphs show mean ± S.E.M for at least 100 μm of the plasma membrane, over two independent experiments. See also *Figure 1—figure supplement 1*.

The following figure supplement is available for figure 1:

**Figure supplement 1.** Controls for specificity.

(*Waheed et al., 2001*; *Möbius et al., 2002*, *2003*; *Kwiatkowska et al., 2014*). The authors of these studies reported that in erythrocytes, lymphoblastoid cells, and platelets, cholesterol was mainly associated with the plasma membrane, MVBs (especially ILVs), and tubular endosomes (*Möbius et al., 2002*). Unfortunately, the reagent they used is no longer available, and no other techniques for EM localisation of cholesterol have been described. Recently, however, theonellamides (TNMs) labeled with fluorescent dyes, such as BODIPY, have been shown to be effective tools for visualizing sterols in fixed cells by fluorescence microscopy (*Nishimura et al., 2013*). Because there are BODIPY antibodies available, we reasoned that BODIPY-conjugated TNM (TNM-BF) might be a suitable reagent for immuno-gold EM localization of cholesterol.

Double labeling fluorescence microscopy with filipin and TNM-BF showed that the two probes have virtually identical patterns in HeLa cells, with labelling particularly concentrated in the juxtanuclear region (*Figure 1B*, *Figure 1—figure supplement 1B*). For electron microscopy, we labelled cryosections of control and BafA1-treated cells with TNM-BF, followed by a commercial rabbit anti-BODIPY antibody and protein A coupled to colloidal gold (*Figure 1C*). In both types of cells, we observed strong labelling of MVBs, with gold particles particularly abundant on the ILVs (upper panels), consistent with previous studies using perfringolysin O (*Möbius et al., 2002*). In control cells, we also saw labelling of the plasma membrane (*Figure 1C*, arrowheads). However, in the BafA1-treated cells, the plasma membrane was virtually devoid of label, although there was label associated with extracellular vesicles (*Figure 1C*, arrows). To look specifically at cell surface cholesterol, we performed pre-embedding labelling. Control and BafA1-treated cells were fixed and labelled with TNM-BF followed by anti-BODIPY without permeabilisation. This method showed even more dramatically that cholesterol is lost from the plasma membrane following BafA1 treatment, and also highlighted the strong labelling of extracellular vesicles (*Figure 1D,E*). As a control for the specificity of labeling, we treated cells with methyl-β-cyclodextrin (MβCD), which extracts cholesterol from the plasma membrane. We found a near-complete loss of surface labeling, although endosomes were still labeled (*Figure 1D,E*, *Figure 1—figure supplement 1C*).

These results are largely in agreement with our previous study, in which we used filipin as a cholesterol probe for light microscopy. In both cases, we found a ~50% loss of surface labeling in BafA1-treated cells. However, in our previous study, we were unable to distinguish between the plasma membrane and extracellular vesicles associated with the cell surface. The present study shows that there is in fact a 15-fold loss of plasma membrane cholesterol, with a concomitant rise in cholesterol-positive extracellular vesicles (*Figure 1E*).

## BafA1 treatment causes an increase in exosome release

Are the cholesterol-rich extracellular vesicles that we observe in BafA1-treated cells in fact exosomes, or could they be plasma membrane-derived vesicles? We addressed this question in several ways. First, exosomes have the same diameter as ILVs, i.e., 30–100 nm diameter, while other types of extracellular vesicles are much more heterogeneous in size, often up to 1 μm in diameter. The BafA1-induced vesicles are indistinguishable in appearance from ILVs (*Figure 2A*, inset left). Second, to find out whether the vesicles come from endosomes, we carried out pulse-chase experiments using BSA conjugated to 5 nm colloidal gold as an endocytic tracer. The cells were allowed to endocytose BSA-gold for 10 min, washed, incubated for either 30 min or 4 hr to chase the gold into endosomes or lysosomes respectively, and then treated for 16 hr with BafA1. *Figure 2A* shows that

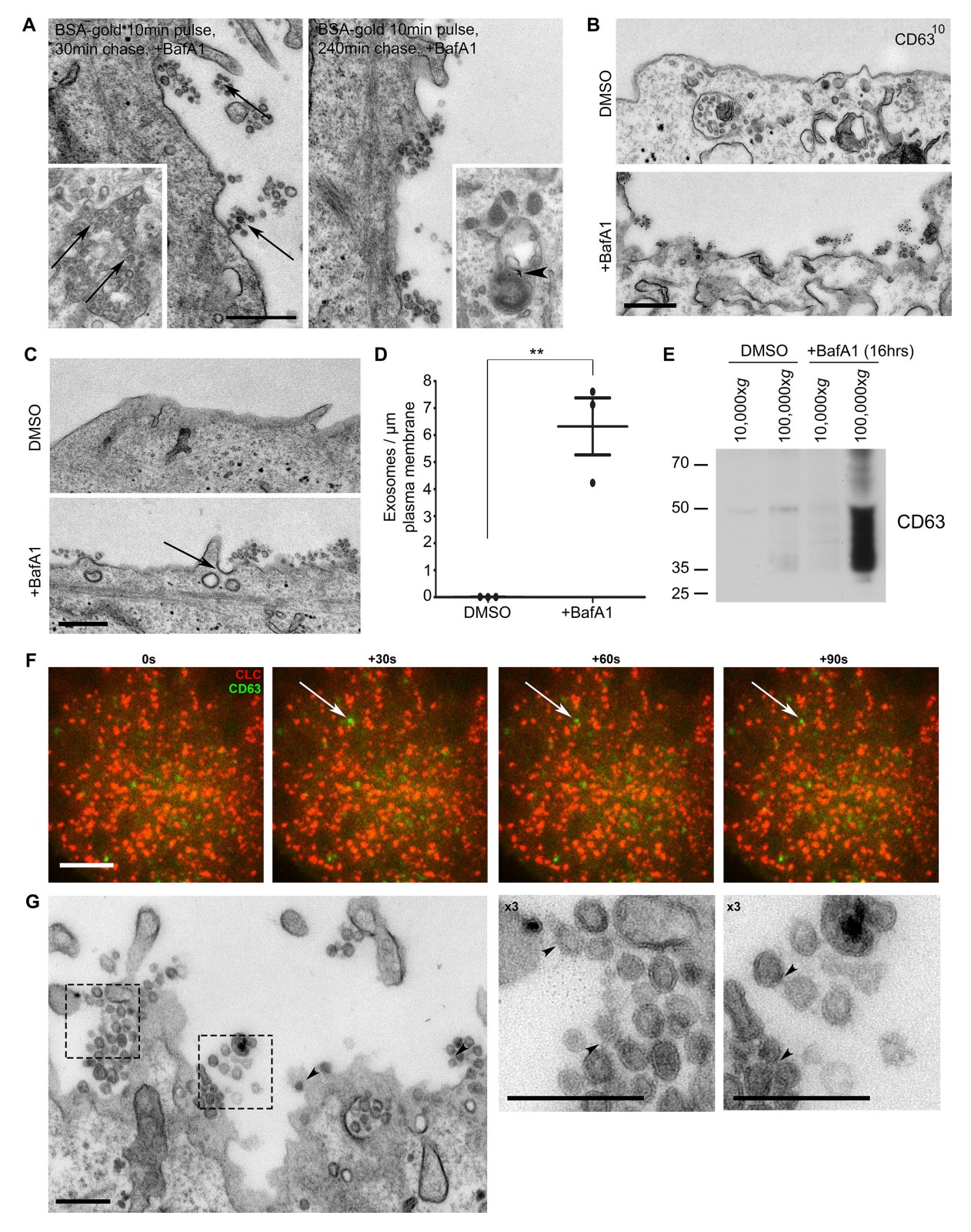

**Figure 2.** The extracellular vesicles that accumulate in BafA1-treated cells are exosomes. (**A**) HeLa cells were incubated with BSA coupled to 5 nm gold for 10 min before being washed several times with PBS to remove any uninternalised label. The cells were then chased in full medium for either 30 min or 4 hr to load BSA-gold into endosomes or lysosomes respectively, then treated with BafA1 (100 nM, 16 hr), fixed, and prepared for conventional EM. Gold could be seen associated with extracellular vesicles from the cells chased for 30 min, but not from the cells chased for 4 hr. Insets: monomeric

*Figure 2 continued on next page*

*Figure 2 continued*

gold could be found within MVBs from cells chased for 30 min (arrows), and aggregated within lysosomes from cells chased for 4 hr (arrowhead). Scale bar: 200 nm. (B) DMSO- or BafA1-treated HeLa cells (100 nM, 16 hr) were surface-labelled with an antibody against the CD63 lumenal domain to identify exosomes. Scale bar: 500 nm. (C) HeLa cells were treated with DMSO or BafA1 (100 nM, 16 hr) before being fixed and prepared for conventional EM. Exosomes were often associated with clathrin-coated pits (arrow). Scale bar: 500 nm (D) The number of exosomes per µm of plasma membrane was quantified. Data shown are means from three independent experiments, ± S.E.M. ***p<0.001 (E) Culture supernatants were collected from mock-treated and BafA1-treated HeLa cells (100 nM, 16 hr), and centrifuged first at 10,000x*g* and then at 100,000x*g*. Western blots of the pellets were probed with anti-CD63. (F) Total internal reflection fluorescence *microscopy* (TIRF) was performed on HeLa cells transiently expressing both CLC-mCherry and CD63-GFP, following BafA1 treatment (100 nM, 16 hr). Representative stills are shown. Scale bar: 10 µm. (G) Conventional EM of BafA1-treated cells (100 nM, 16 hr), with arrowheads indicating proteinacious material on exosomes. Scale bars: 500 nm. See also *Figure 2—figure supplement 1*, *2*.

The following figure supplements are available for figure 2:

**Figure supplement 1.** Prolonged treatment with BafA1 induces exosome release.

**Figure supplement 2.** Exosome-enriched preparations probed with antibodies against various extracellular vesicle markers.

gold could be observed in association with extracellular vesicles in the cells that had been chased for 30 min, but not in the cells that had been chased for 4 hr, indicating that the vesicles are endosomal in origin. Third, we labelled non-permeabilised cells with an antibody against CD63, which has been shown to be enriched on both ILVs and exosomes (*Pols and Klumperman, 2009*), and saw strong labelling of the extracellular vesicles in the BafA1-treated cells, but little or no surface labelling in control cells (*Figure 2B*). Fourth, as shown in *Figure 1*, both the extracellular vesicles and the ILVs of BafA1-treated cells are enriched in cholesterol, while cholesterol is largely absent from the plasma membrane. Together, these observations indicate that the extracellular vesicles that accumulate in BafA1-treated cells are indeed exosomes.

When we quantified the number of exosomes at the plasma membrane, we found that they were almost non-existent in control HeLa cells, but were very abundant (~5–8 exosomes per µm plasma membrane) after prolonged treatment with BafA1 (*Figure 2C,D*, *Figure 2—figure supplement 1*). We also used a biochemical approach to investigate exosome release, by collecting culture medium from control and BafA1-treated cells and centrifuging it first at 10,000x*g* (which enriches for larger particles like plasma membrane-derived vesicles and apoptotic bodies) and then at 100,000x*g* (which enriches for exosomes). Western blots probed with an antibody against CD63 showed that the marker was barely detectable in the fractions from control cells, but extremely abundant in the 100,000x*g* pellet from BafA1-treated cells (*Figure 2E*), consistent with our EM observations. We also probed the exosome-enriched preparations for other extracellular vesicle markers, including Alix, Tsg101, and CD9 (*Figure 2—figure supplement 2*), and in most cases, we saw at least a slight effect of BafA1 treatment. However, Western blotting is not the most precise way of quantifying differences in protein concentration, and in future we intend to use mass spectrometry for more accurate and comprehensive analyses.

The BafA1-induced exosomes are often in close proximity to the non-constricted clathrin-coated pits that we described in our previous study (e.g., see the arrow in *Figure 2C*), and we speculated that there might be a temporal relationship between exosome release and clathrin-coated pit formation, with frequent fusion events followed by a burst of clathrin recruitment. To investigate this relationship further, we cotransfected cells with GFP-tagged CD63 and mCherry-tagged clathrin light chain. Live-cell TIRF imaging of BafA1-treated cells showed that under these conditions, clathrin is in fact very static and fusion events are relatively rare. However, when fusions do occur, the CD63-GFP signal persists rather than diffusing into the medium, and the ventral surface of the cell is decorated with stable CD63-GFP puncta of varying intensities (*Figure 2F*). The frequency of exosomes in thin sections of BafA1-treated cells, compared with the rarity of CD63-GFP fusion events, indicates that the exosomes are somehow tethered to the plasma membrane, rather than released as diffusible vesicles. We have previously hypothesised that ILVs are held together inside endosomes by an unknown material that can be observed by electron microscopy (*Edgar et al., 2014*). Careful analysis

showed that exosomes released from BafA1-treated cells display a similar material, which may cross-link them together (*Figure 2G*).

## Tetherin links exosomes to the plasma membrane

One candidate for a protein that might attach exosomes both to the plasma membrane and to each other is tetherin, also called Bst2, CD317, and HM1.24. Tetherin is an interferon-inducible Type II transmembrane protein with a GPI anchor at its C terminus. It acts to inhibit the spread of certain enveloped viruses, including HIV, by cross-linking the virions and holding them together at the plasma membrane (*Neil et al., 2008*). We speculated that tetherin might act in a similar manner on exosomal vesicles (*Figure 3A*).

Immunofluorescence labelling of permeabilised cells showed that endogenous tetherin in HeLa cells is localised to a juxtanuclear compartment, both under control conditions and after BafA1 treatment (*Figure 3B*). This tetherin labelling colocalised with CD63 labelling, indicating that the juxtanuclear compartment is endosomal (*Figure 3—figure supplement 1A*). In non-permeabilised cells, where the antibody was only able to access the cell surface, there was relatively little tetherin labelling under control conditions, but BafA1 treatment caused an increase in surface tetherin labelling (*Figure 3B*). Again, there was excellent colocalisation between tetherin and CD63 (*Figure 3—figure supplement 1B*). Pre-embedding EM labelling of BafA1-treated cells revealed that this surface labelling was concentrated on exosomes (*Figure 3C*, *Figure 3—figure supplement 2*).

However, the presence of tetherin on exosomes after BafA1 treatment does not necessarily mean that tetherin stabilises the association of exosomes with the plasma membrane. Tetherin is known to be trafficked through the endocytic pathway (*Neil et al., 2008*; *Habermann et al., 2010*), and thus would be released following endosome-plasma membrane fusion even if it were not playing an active role in exosome aggregation. In order to determine whether tetherin actually holds exosomes together, we used CRISPR/Cas9-mediated genome editing to knock out the tetherin gene in HeLa cells. Clonal cell lines were assayed by Western blotting, and a cell line in which tetherin was no longer expressed was selected for further studies (*Figure 3D*).

To investigate the effect of tetherin loss on exosome distribution, EM was performed on control HeLa cells and on our knockout cell line, following BafA1 treatment. We observed fewer exosomes in the tetherin knockout cells, and those that we did find appeared to be less aggregated (*Figure 3E*). Quantification of three independent experiments revealed a ~4-fold decrease in exosomes at the plasma membrane in the tetherin knockout cells compared with controls (*Figure 3F*). We also probed Western blots of exosome-enriched preparations from the culture supernatant of control and tetherin knockout cells, treated with or without BafA1, to see whether the decrease in exosomes at the plasma membrane could be correlated with an increase in the discharge of exosomes into the medium. There was a weak signal for CD63 in the knockout cells even without BafA1 treatment, and a strong increase in the signal from BafA1-treated tetherin knockout cells compared with BafA1-treated wild-type cells (*Figure 3G*). Quantification of the signal from three experiments showed a ~4-fold increase, in agreement with our EM results. We also probed for three other extracellular vesicle-associated proteins, Alix, Tsg101, and CD9 (*Figure 3—figure supplement 3*). We found that the tetherin knockout appeared to have a weak effect on Alix and Tsg101, and a stronger effect on CD9. Again, it will be important to confirm and extend these observations by mass spectrometry.

To obtain a three-dimensional view of exosomes associated with the plasma membrane, we performed scanning electron microscopy on wild-type and tetherin knockout cells (*Figure 4*). In untreated wild-type cells, the plasma membrane was essentially devoid of extracellular vesicles, supporting our TEM analysis. Following BafA1-treatment, wild-type cells displayed extracellular vesicles that were not randomly distributed, but rather appeared in clusters. There were also extracellular vesicles associated with the plasma membrane of tetherin knockout BafA1-treated cells, but they were less abundant and the clusters were generally smaller. By combining scanning EM with immunogold labeling, we were able to visualize the surface distribution of both cholesterol (using TNF-BF) (*Figure 5A*) and tetherin (*Figure 5B*). Both were found to be scattered evenly over the surface of untreated cells, but to localize to exosome clusters after BafA1 treatment.

Tetherin has been proposed to function by using its transmembrane domain and its GPI anchor to insert itself into two apposing membranes, the plasma membrane and the viral envelope, and removal of the GPI anchor has been shown to abolish its ability to sequester viruses (*Neil et al.,*

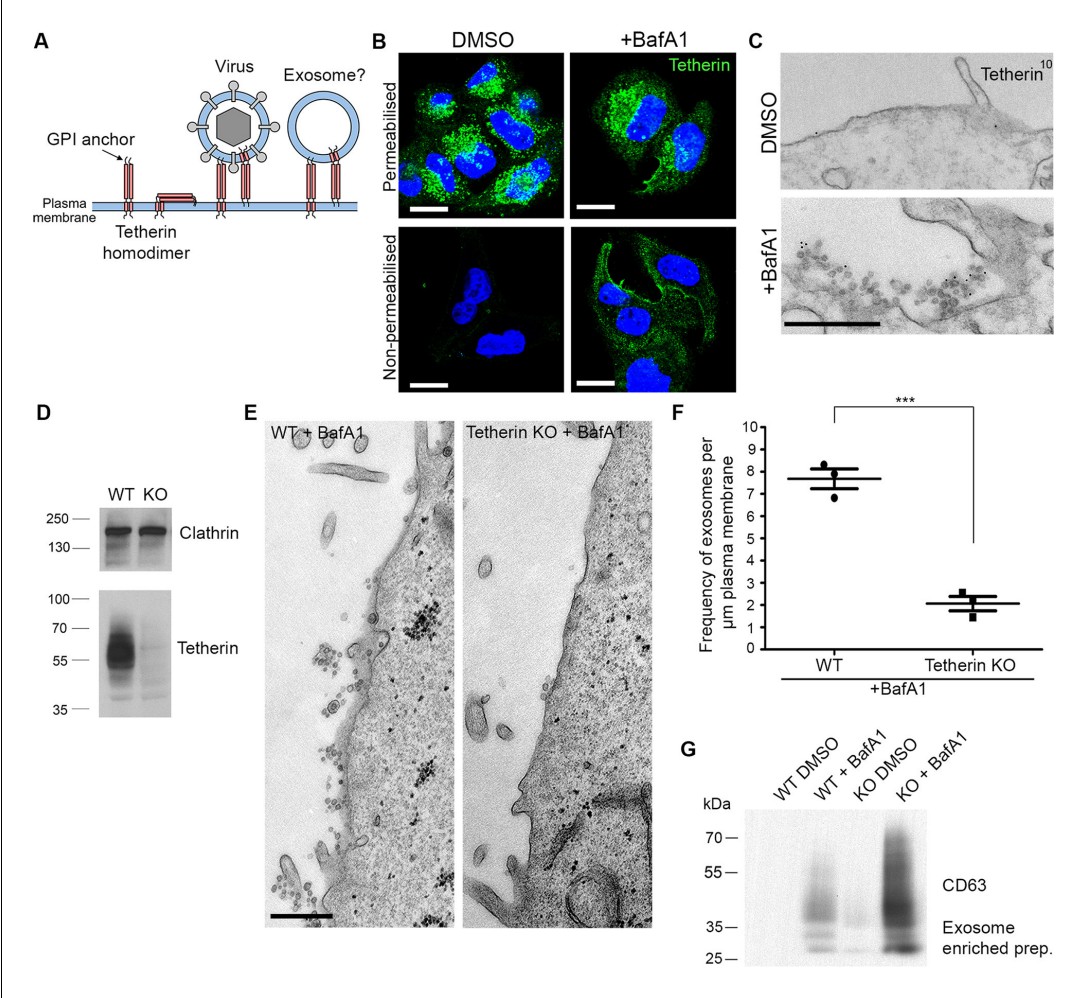

**Figure 3.** Tetherin localises to exosomes and facilitates exosome tethering. (**A**) Schematic diagram of tetherin. (**B**) Mock-treated or BafA1-treated HeLa cells (100 nM, 16 hr) were either permeabilised or left intact and stained with an anti-tetherin antibody. Scale bars: 20 μm. (**C**) Mock-treated or BafA1-treated HeLa cells (100 nM, 16 hr) were surface-labelled using an anti-tetherin antibody followed by 10 nm protein A-gold. Scale bar: 500 nm. (**D**) The tetherin gene was knocked out using CRISPR/Cas9, and the loss of tetherin in a clonal population was confirmed by Western blotting. (**E**) Wild-type or tetherin-knockout HeLa cells were treated with BafA1 (100 nM, 16 hr) and processed for EM to analyse exosome frequency. Scale bar: 500 nm. (**F**) The frequency of exosomes per μm plasma membrane in BafA1-treated (100 nM, 16 hr) wild-type or tetherin-knockout cells was calculated. The mean ± S.E.M are shown from three independent experiments. ***p<0.001. (**G**) Exosome-enriched preparations were generated from the culture supernatants of both wild-type and tetherin-knockout cells, either with or without BafA1-treatment (100 nM, 16 hr), and Western blots were probed with anti-CD63. See also *Figure 3—figure supplements 1*, *2*, *3*.

The following figure supplements are available for figure 3:

**Figure supplement 1.** Tetherin and CD63 colocalise in mock-treated and BafA1-treated cells.

**Figure supplement 2.** Quantification of surface labeling.

**Figure supplement 3.** Exosome-enriched preparations probed with antibodies against other extracellular vesicle markers.

*2008*) (see *Figure 3A*). To investigate whether tetherin uses the same strategy to attach exosomes to the plasma membrane and to each other, we transiently transfected tetherin knockout HeLa cells with either wild-type tetherin or a △GPI mutant, both tagged with HA. Expression of the two constructs was confirmed by probing Western blots with an antibody against the HA tag (*Figure 6—figure supplement 1*). The wild-type tetherin construct was found to rescue the knockout phenotype,

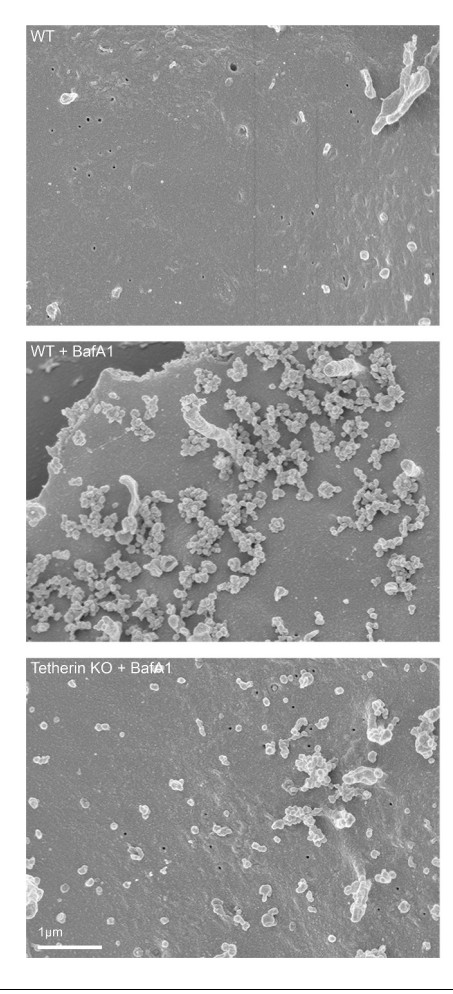

**Figure 4.** Scanning electron microscopy reveals clustering of exosomes on the cell surface. Wild-type and tetherin-knockout cells were treated with or without BafA1 (100 nM, 16 hr) before being fixed and prepared for scanning electron microscopy. Scale bar: 1 μm.

and the clusters of exosomes on the cell surface were heavily decorated with anti-HA. In contrast, the △GPI mutant failed to rescue the phenotype: exosomes were scarce, and the construct localised to the entire plasma membrane (*Figure 6A*). In both cases, however, there was strong labeling of the lumen of MVBs, indicating that tetherin does not need its GPI anchor to be sorted into ILVs (*Figure 6B*).

Finally, to extend our findings to a more physiological system and to ensure that the tethering we observe in HeLa cells is not due to endosome acidification defects, we investigated EBV-transformed B cell lines. These cells have been shown to generate exosomes at steady state without a stimulus (*Raposo et al., 1996*). We observed not only clustering of exosomes by conventional EM (*Figure 7A*), but also tetherin labeling on the clustered exosomes and on ILVs by cryo immuno EM (*Figure 7B*).

## Discussion

In the present study, we have shown that V-ATPase is a key regulator of both cholesterol trafficking and endosome fate, with a dramatic increase in plasma membrane-associated exosomes when V-ATPase is inhibited by BafA1. We have also shown that these exosomes are attached to the plasma membrane and to each other by the antiviral protein tetherin.

Our findings on cholesterol trafficking absolutely depended upon being able to localise cholesterol at the ultrastructural level, using TNM-BF as a probe. We found that there is practically no cholesterol associated with the plasma membrane after V-ATPase inhibition; instead, the cholesterol is associated with extracellular vesicles, a distinction that was not apparent by light microscopy. There are still a number of unanswered questions about the connection between V-ATPase and cholesterol, including precisely how endosomal pH affects cholesterol trafficking. The availability of TNM-BF as an EM-compatible cholesterol probe should allow these and other questions to be addressed.

Exosomes are not the only type of extracellular vesicles produced by cells; other types include microvesicles, produced by plasma membrane shedding, and apoptotic bodies, produced by a controlled 'explosion' of cells undergoing apoptosis. Hence, there is some confusion in the literature because not all studies purporting to be about exosomes actually distinguished between different types of extracellular vesicles (*Raposo and Stoorvogel, 2013*). We used several criteria to confirm the identity of the vesicles we observed, including similarity in size to ILVs (the other types of vesicles are generally larger and more heterogeneous), the presence of a previously endocytosed tracer, and enrichment of cholesterol and CD63. The connection between V-ATPase inhibition and exosome production is at present unclear, and indeed, little is known about the machinery involved in endosome-plasma membrane fusion in general, in part because such fusion events are rare in most cell types. Our finding that endosome-plasma membrane fusion can be induced by blocking endosome acidification should facilitate the discovery of some of the players in this event.

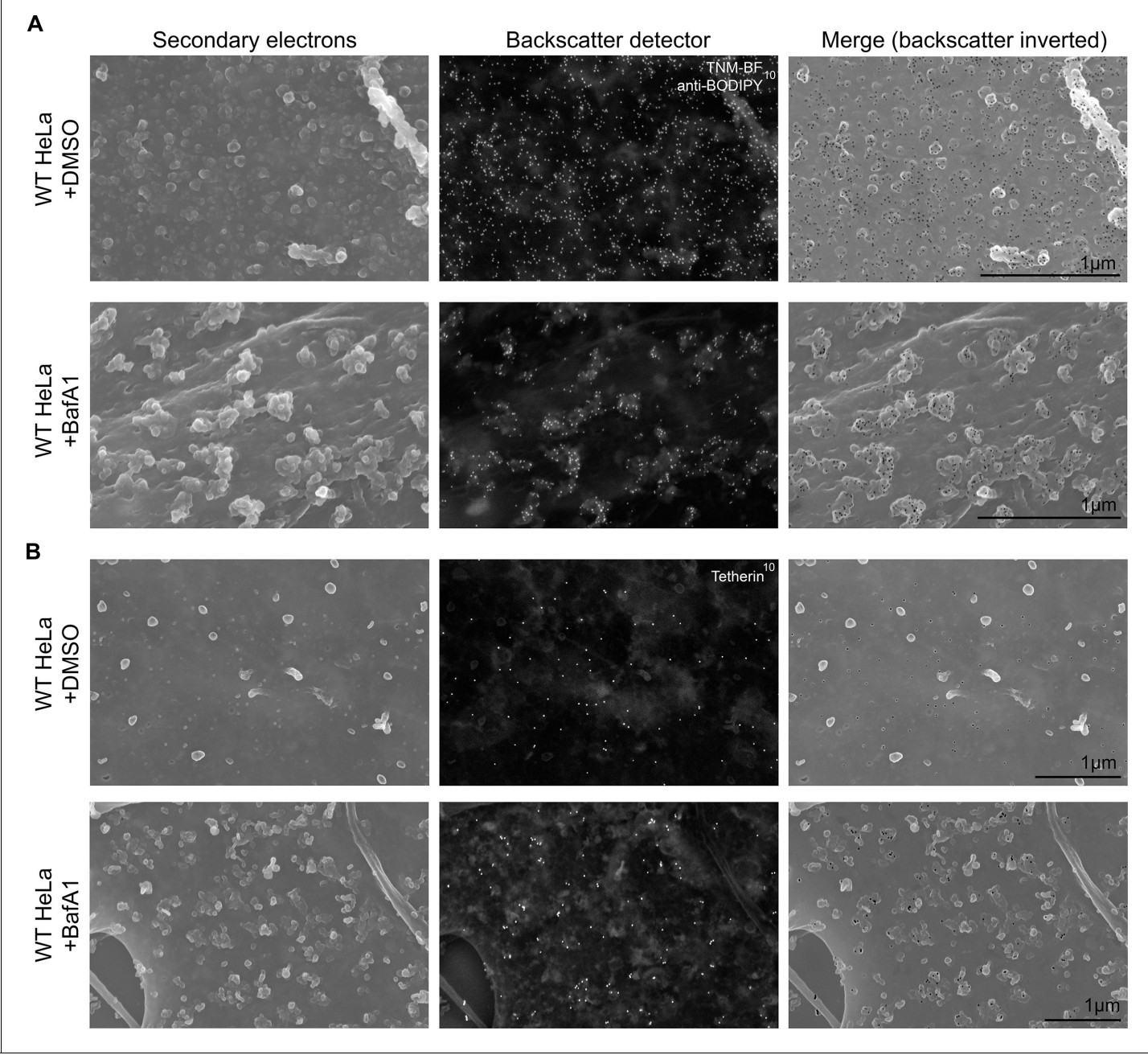

**Figure 5.** Immuno-SEM reveals the localization of cholesterol and tetherin in 3D. HeLa cells treated with or without BafA1 (100 nM, 16 hr) were fixed, surface-labelled, and viewed by scanning electron microscopy. The cells in **A** were probed for cholesterol and the cells in **B** were probed for tetherin. Small 'bumps' observed on the plasma membrane probably represent surface-bound antibody complexes. Scale bars: 1 μm.

Although exosomes are generally thought of as free extracellular vesicles, our live cell imaging results and EM images, together with published micrographs of B cell exosomes (*Raposo et al., 1996*; *Simons and Raposo, 2009*), suggested to us that exosomes can have an alternative fate: to stay closely associated with the cell that produced them. We decided to investigate the potential role of tetherin in this association, because we were struck by the similarity between our images of exosomes in BafA1-treated cells, and published images of HIV-1 budding from cells in which tetherin was overexpressed and the tetherin antagonist, Vpu, deleted from the viral genome (*Neil et al., 2008*). We discovered that both endogenous and tagged tetherin localise to exosomes, that

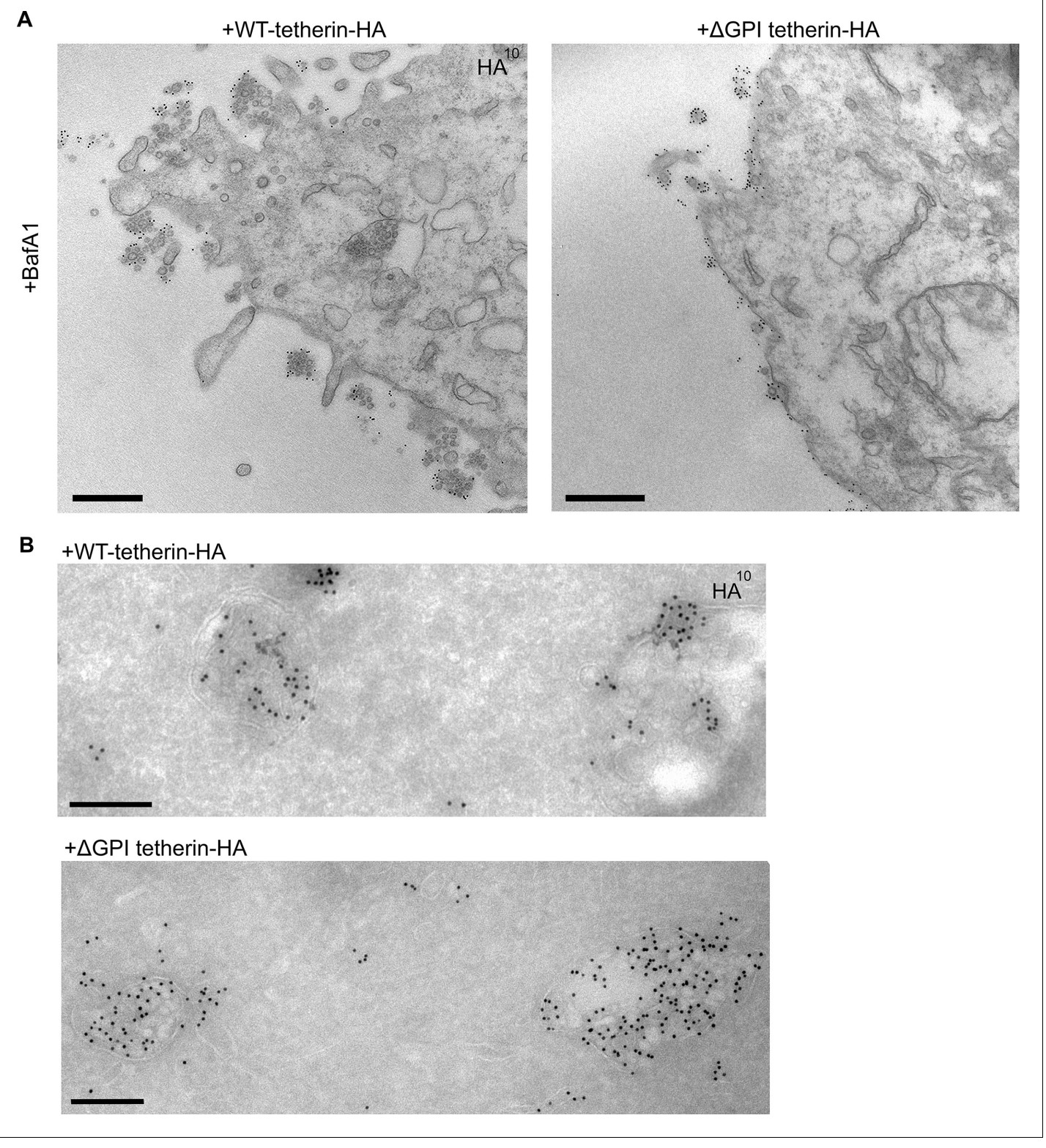

**Figure 6.** Exosome tethering can be rescued by WT-tetherin, but not by ΔGPI-tetherin. (**A**) Tetherin-knockout cells were transiently transfected with WT-tetherin-HA or ΔGPI-tetherin-HA, treated with BafA1 (100 nM, 16 hr), and prepared for surface labelling EM. Cells expressing tetherin-HA constructs were identified by surface gold labelling using an anti-HA antibody. Scale bars: 500 nm. (**B**) Tetherin-knockout HeLa cells were transiently transfected with either WT-tetherin-HA or ΔGPI-tetherin-HA constructs. Cryosections of both were immunolabelled with anti-HA antibodies, revealing localisation to the ILVs of MVBs. Scale bars: 200 nm. See also *Figure 6—figure supplement 1*.

*Figure 6 continued on next page*

*Figure 6 continued*

The following figure supplement is available for figure 6:

**Figure supplement 1.** Western blotting of tetherin rescue cells.

knocking out tetherin strongly reduces the number of exosomes associated with the plasma membrane with a concomitant increase in exosomes released into the medium, and that this phenotype can be rescued by wild-type tetherin but not by a tetherin construct lacking a GPI anchor, which is also incapable of tethering virus particles. One difference between exosomes and viruses is that whereas overexpression of tetherin in virally infected cells leads to a massive accumulation of virions at the plasma membrane (*Neil et al., 2008*), overexpression of tetherin in BafA1-treated cells did not produce any marked increase in exosome frequency. This may be because the unit number of exosomes is limited by the number of ILVs per endosome, whereas cells infected with a virus such as HIV-1 are programmed to bud as many viruses as possible from the plasma membrane, and so are less confined by space and membrane availability.

A single endosome can generate ILVs via distinct mechanisms, resulting in a heterogeneous population of vesicles (*Edgar et al., 2014*). If the endosome then fuses with the plasma membrane, all of the vesicles will be discharged; however, the ultimate fate of each vesicle may depend upon whether or not it contains tetherin. The presence of a GPI anchor suggests that tetherin may preferentially insert into the most cholesterol-rich vesicles; however, in order for tetherin to tether vesicles to each other as well as to the plasma membrane, it must be able to insert into the vesicles both via its GPI anchor and via its transmembrane domain (see *Figure 3A*) (*Venkatesh and Bieniasz, 2013*). In any case, a non-homogenous distribution of tetherin would provide the cell with a mechanism for keeping some vesicles attached the cell surface while releasing others into the extracellular matrix or fluid. In addition, expression of tetherin is tightly regulated. It is most highly expressed in cells of the immune system (*Ishikawa et al., 1995*; *Goto et al., 1994*; *Blasius et al., 2006*) but even in cells where it is normally undetectable, its expression can be dramatically upregulated by interferon (*Goto et al., 1994*; *Blasius et al., 2006*). This tight control of tetherin expression may provide a means whereby the cell can regulate the fate of its exosomes, and thus control whether the exosomes engage in local interactions (e.g., antigen presentation via an immunological synapse [*Raposo et al., 1996*]) or long-distance interactions (e.g., supporting or inhibiting cell migration [*Sung et al., 2015*]). Supporting this notion, tetherin has recently been shown to promote dendritic cell activation and antigen presentation via MHC class II (*Li et al., 2016*), a pathway that has been shown to be mediated by exosomes (*Raposo et al., 1996*).

There is currently great interest in exosomes as vehicles for intercellular communication, and clinical studies are underway to exploit them both as diagnostic tools and as potential therapeutic agents. However, as several recent review articles have pointed out, we still don't really know what exosomes contain, because there is no way of specifically purifying them. In previous work from our lab on clathrin-coated vesicles (CCVs), we have shown that even with an impure preparation, we can deduce the protein composition by comparative proteomics (e.g., by preparing the CCV fraction from control vs. clathrin-depleted cells [*Borner et al., 2012*]). The present study demonstrates that one can specifically enrich or deplete exosomes from the extracellular fluid by manipulating endosomal pH and/or tetherin levels. Thus, by comparing exosome-enriched fractions prepared from cells treated in different ways, it should be possible to determine which proteins in the fraction are actually exosome components and which are contaminants.

Exosomes have been proposed to act as shuttles for the spread and propagation of prions (*Fevrier et al., 2004*), beta amyloid (*Rajendran et al., 2006*), and alpha-synuclein (*Danzer et al., 2012*), all of which can lead to severe neurodegenerative disorders. A number of enveloped viruses, including HIV-1, have developed ways of keeping tetherin off the plasma membrane, and one of the consequences of this downregulation may be that exosome interactions are altered in infected cells. Interestingly, HIV-1-infected patients display an increased propensity to develop neurodegenerative disorders (HAND – HIV-1 Associated Neurodegeneration), as well as increased beta amyloid (*Achim et al., 2009*) and alpha-synuclein deposition (*Khanlou et al., 2009*). Exosomes have also

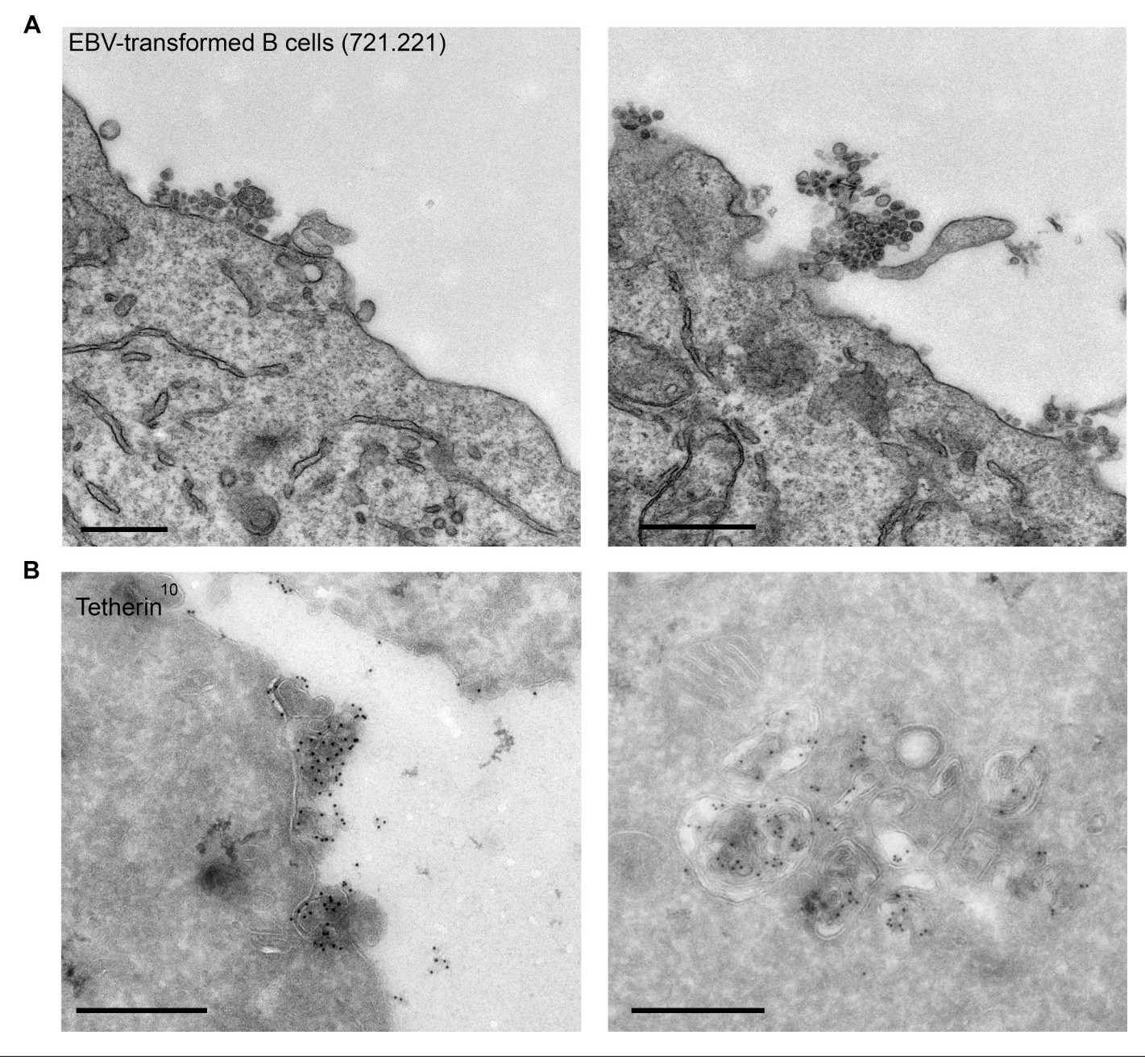

**Figure 7.** EBV-transformed B cells display clusters of tetherin-positive exosomes. (**A**) 721.221 EBV-transformed B cells were prepared for conventional EM. Clusters of exosomes can be seen at the cell surface. Scale bars: 500 nm. (**B**) Immuno-EM reveals that the clustered exosomes and the ILVs of MVBs are positive for tetherin labeling. Scale bars: 500 nm.

been shown to play a pathological role in cancer. Tumor cells frequently shed excessive amounts of exosomes, and these are thought to promote the growth, survival, and metastasis of the tumor in a number of ways (*Zhang and Wang, 2015*; *Zhang et al., 2015*; *Ono et al., 2014*; *Melo et al., 2014*; *Anastasiadou and Slack, 2014*; *Zhao et al., 2016*). Interferon-α is sometimes given as a chemotherapeutic drug, and it is tempting to speculate that one of its mechanisms of action may be to prevent exosome shedding, by upregulating tetherin expression in the tumor cells. Thus, the importance of tetherin in health and disease is likely to extend well beyond its role as an antiviral protein.

# Materials and methods

## Cell culture

HeLa cells were a gift from Paul Lehner (University of Cambridge, UK) and were cultured in DMEM supplemented with 10% fetal bovine serum, L-glutamine, and *Penicillin/Streptomycin,* in 5% $CO_2$ at 37°C. The CD63-GFP construct was a gift from Paul Luzio (University of Cambridge, UK), the CLC-mCherry construct was made in-house, and the tetherin-HA constructs were as described (*Billcliff et al., 2013*). Transient transfections were performed using TransitIT-HeLa MONSTER (Mirus).

Tetherin knockout HeLa cells were generated using the CRISPR/Cas9 method. Guide RNAs targeting a sequence within the first exon of the tetherin gene (5'-GCTCCTGATCATCGTGATTC**TGG**) were cloned into pX330 (Addgene plasmid #42230). Monoclonal HeLa cell lines were generated and assayed for expression of the tetherin protein by immunofluorescence and Western blotting. An apparent tetherin negative line was selected, and the absence of wild-type tetherin sequence was confirmed by PCR-cloning and Sanger sequencing of the targeted region.

The EBV-transformed B cell line (721.221) was a gift from Gillian Griffiths (University of Cambridge, UK) and were cultured in RPMI supplemented with 10% fetal bovine serum, L-glutamine, and Penicillin/Streptomycin, in 5% $CO_2$ at 37°C.

All cells were free of mycoplasma.

## Antibodies

Antibodies used for immunofluorescence included a mouse monoclonal anti-CD63, IB5 (a gift from Mark Marsh, UCL, UK), rabbit polyclonal anti-BODIPY (A-5770, Molecular Probes - RRID:AB_2536193), rabbit polyclonal anti-Bst2/tetherin (NIH AIDS reagent program), and a mouse monoclonal antibody against tetherin, HM1.24 (*Goto et al., 1994*). For Western blotting, the following antibodies were used: a mouse polyclonal anti-tetherin antibody, B02P (Abnova, Taiwan - RRID:AB_1137604), a mouse monoclonal antibody against the HA tag (Covance, 16B12 - RRID:AB_10064220), a goat polyclonal anti-EF2 antibody (Santa Cruz, C-14 - RRID:AB_640040), a rabbit polyclonal anti-Alix antibody (a gift from Harald Stenmark, University of Oslo, Norway), a mouse monoclonal anti-Tsg101 antibody (GeneTex - RRID:AB_373239), a rabbit monoclonal anti-CD9 antibody (Abcam, ab92726 - RRID:AB_10561589) and a rabbit polyclonal anti-clathrin antibody (made in-house [*Simpson et al., 1996*]). A rabbit anti-mouse IgG antibody (Dako) was used as a bridging antibody for immuno-EM.

## Reagents

Magic Red-Cathepsin B (ICT938, BioRad) and Lysotracker Red DND-99 (L7528, ThermoFisher) were used for live cell imaging. Filipin complex (F9765, Sigma) was dissolved in DMSO, and used at a 25 µg/ml. TNM-BF was synthesized as described previously (*Nishimura et al., 2010*). Protocols for labeling are detailed below. Methyl-β-cyclodextrin (C4951, Sigma-Aldrich) was dissolved in water and used at a final concentration of 10 mM for 30 min. BafA1 (B1793, Sigma-Aldrich) was dissolved in DMSO and used at a final concentration of 100 nM. Control experiments were performed using equal volumes of DMSO. Cell viability with and without bafilomycin treatment was assessed by trypan blue exclusion staining, and in both cases it was found to be >98%.

## Immunofluorescence

Cells were fixed with 4% PFA/PBS and permeabilised using 0.1% saponin. Blocking and subsequent steps were performed with 1% BSA, 0.01% saponin in PBS. Cells were mounted on slides with mounting medium containing DAPI (Invitrogen).

## Filipin and TNM-BF double labeling immunofluorescence

Cells were fixed with 4% PFA. They were initially stained with filipin, before being permeabilised with 0.1% saponin, blocked with 1% BSA, 0.01% saponin in PBS, and stained with 1 µM TNM-BF for 30 min on ice. The cells were washed with PBS before being mounted on slides with mounting medium (Invitrogen), and visualized using a Zeiss 880 confocal microscope and 63X objective.

## Western blotting

Western blotting for tetherin and for CD63 was performed in non-reducing conditions and tetherin blots using a sample buffer without bromophenol blue (10% SDS, 15% glycerol, 0.2 M Tris-HCl pH 6.8) as previously described (*Giese and Marsh, 2014*). Other Western blots were performed according to standard protocols. Signals were quantified using ImageJ. Exosome-enriched preparations were generated from cell culture supernatants as previously described (*Théry et al., 2006*). Cells were grown in equal numbers on 245 mm x 245 mm dishes in 60 ml of media. Supernatants were collected and cells removed by a low speed spin (300x*g*, 10 min). Pellets were discarded and supernatants were then spun to remove large cell debris (2000x*g*, 10 min). Again pellets were discarded and supernatants were ultracentrifuged (Type 70 Ti Rotor, Beckman Coulter) to remove smaller cell debris (10,000x*g*, 30 min). Pellets were retained for controls and supernatants were again collected and ultracentrifuged (Type 70 Ti Rotor, Beckman Coulter) at 100,000x*g* for 70 min. Pellets were washed in PBS and repelleted at 100,000x*g* for 70 min (Type 70 Ti Rotor, Beckman Coulter) to give exosome-enriched fractions. Exosome-enriched pellets were resuspended in 100 µl sample buffer and equal volumes loaded for Western blotting. Quantification of Western blotting was performed using ImageJ.

## Cryo-immuno electron microscopy

Cells were fixed with 4% PFA, 0.2% glutaraldehyde, 0.1 M phosphate buffer (pH 7.4). They were then pelleted in 12% gelatine and 70 nm sections cut at −120°C with a Leica ultracut UCT ultramicrotome. Sections were labelled with 10 nm protein-A gold (Utrecht university, the Netherlands) and stained as previously described (*Slot et al., 1991*).

## Cryo-immuno electron microscopy to localise cholesterol

Control or BafA1-treated HeLa cells (100 nM, 16 hr) were fixed with 4% PFA, 0.2% glutaraldehyde, 0.1 M phosphate buffer (pH 7.4). Cells were pelleted in 12% gelatine and 70 nm sections cut at −120°C with a Leica ultracut UCT ultramicrotome. Sections were picked up in 2.3 M sucrose and grids were blocked with 2% gelatin. Prior to gold labeling, grids were washed with 15 mM glycine and then washed twice with PBS. Sections were incubated on drops of 1 µM TNM-BF for 30 min on ice, washed, blocked with 1% BSA in PBS, and then incubated with a rabbit polyclonal anti-BODPIY antibody (Molecular Probes, Invitrogen, UK). Sections were labelled with 10 nm protein-A gold (Utrecht university, the Netherlands) and stained as previously described (*Slot et al., 1991*).

## Conventional electron microscopy

HeLa cells were grown on 13 mm diameter Thermanox coverslips (Nalge Nunc International, Rochester, NY,) and briefly fixed in their culture medium with an equal volume of double strength fixative (4% PFA, 5% glutaraldehyde, 0.1 M cacodylate buffer pH 7.4), before being exchanged for single strength fixative (2% PFA, 2.5% glutaraldehyde in 0.1 M cacodylate buffer pH 7.4). EBV-transformed B cells were briefly fixed in a double strength fixative (4% PFA, 5% glutaraldehyde, 0.1 M cacodylate buffer pH 7.4) at an equal volume to their culture medium before being spun into a pellet. Cells were post-fixed with 1% osmium tetroxide and incubated with 1% tannic acid to enhance contrast. They were dehydrated using increasing concentrations of ethanol before being embedded onto EPON stubs (coverslips) or EPON-filled molds (pellets), and cured overnight at 65°C. Coverslips were removed using a heat-block. Ultrathin 70 nm conventional sections were cut using a diamond knife mounted to a Reichart ultracut S ultramicrotome. Sections were stained with lead citrate before being observed on a FEI Tecnai transmission electron microscope (FEI, the Netherlands) at an operating voltage of 80 kV.

## Correlative Light and Electron Microscopy (CLEM)

Cells were grown on gridded glass coverslips (MatTek Corporation, Ashland, MA). Immunofluorescence staining was performed as described above, following which cells were refixed with 2% PFA, 2.5% glutaraldehyde, 0.1 M cacodylate and processed for conventional EM. Resin stubs were inverted over areas of interest and baked overnight. Coverslips were subsequently removed using liquid nitrogen.

## Pre-embedding surface labeling of cell surface/exosomal antigens

Cells were grown on Thermanox coverslips (Nalge Nunc International, Rochester, NY) and briefly fixed with 4% PFA/0.1 M cacodylate at a 1:1 ratio with culture medium. After 2 min, fixative was replaced with fresh 2% PFA/0.1 M cacodylate. Cells were blocked with 1% BSA, 0.1% Aurion BSA (Aurion, the Netherlands) before being incubated with mouse monoclonal antibodies against lumenal/extracellular antigens for 1 hr at room temperature, followed by incubation with a rabbit anti-mouse secondary antibody (Dako). The cells were then labeled with 10 nm protein-A gold (Utrecht university, the Netherlands), and re-fixed with 2% PFA, 2.5% glutaraldehyde / 0.1 M cacodylate, before being post-fixed with 1% osmium tetroxide and processed for conventional resin embedding as above. For cell surface cholesterol labeling, cells were fixed as above. Coverslips were then incubated with 1 µM TNM-BF for 30 min on ice, washed, and incubated with a rabbit polyclonal anti-BODIPY antibody followed by 10 nm protein-A gold as above.

## BSA-gold feeding

BSA conjugated to 5 nm colloidal gold was prepared as previously described (*Slot and Geuze, 1985*). Cells were incubated with BSA-gold for 10 min, followed by five washes with PBS to remove surface BSA-gold. The cells were then chased in fresh medium for either 30 min or 4 hr to load endosomes or lysosomes respectively, before addition of BafA1 (100 nM, 16 hr). Cells were then fixed for conventional electron microscopy (as above).

## Quantification of exosomes per unit plasma membrane

Exosomes were counted manually and lengths of plasma membrane calculated using ImageJ. Five consecutive photographs along random portions of plasma membrane were imaged, from at least five random cells – omitting areas where cells were within 5 µm of another cell. Only exosomes within 500 nm of plasma membrane were quantified. Three independent experiments were performed.

## Quantification of surface labeling immuno gold density

The number of gold particles associated with plasma membrane or extracellular vesicles per image were counted. The length of plasma membrane (µm) per image was then calculated using ImageJ and the number of gold particles per µm were calculated for each image. Over 100 µm total plasma membrane was quantified per condition, over two separate experiments.

## Scanning EM

Cells grown on glass coverslips were fixed with 2% PFA/ 2.5% glutaralydehyde, 0.1 M cacodylate and secondarily fixed with 1% osmium. After critical point drying, the cells were coated with 15 nm of carbon using a Quorum Q150 carbon coater and imaged with a FEI Verious 460 SEM.

## Immuno-SEM

Cells were grown on glass coverslips and fixed with 4% PFA in PBS. Coverslips were labeled with gold as described above (pre-embedding surface labeling of cell surface/exosomal antigens). After gold labeling, cells were refixed with 2% PFA, 2.5% glutaraldehyde, and sputter coated with a thin layer of carbon using a Quorum Q150 carbon coater. SEM images were collected using an FEI Various 640 SEM. A secondary detector was used to acquire cell surface images, immediately followed by acquisition using a backscatter detector to detect gold particles.

## Acknowledgements

We would like to thank Paul Luzio (CIMR, University of Cambridge) for discussions and critical reading of the manuscript, Jeremy Skepper (Cambridge Advanced Imaging Centre, University of Cambridge) for aiding SEM experiments, Nick Bright (CIMR, University of Cambridge) for advice on EM techniques, and all members of the Robinson lab for helpful discussions. This work was supported by grants from the Wellcome Trust: 086598 (to MSR), 100140 (Wellcome Trust Strategic Award) and 093026 (for FEI Tecnai G2 Spirit BioTWIN transmission EM). The authors declare no conflicts of interest.

## Additional information

### Funding

| Funder | Grant reference number | Author |
| --- | --- | --- |
| Wellcome Trust | 086598 | James R Edgar<br>Paul T Manna<br>Margaret S Robinson |
| Wellcome Trust | 100140 | James R Edgar<br>Paul T Manna<br>Margaret S Robinson |
| Wellcome Trust | 093026 | James R Edgar<br>Paul T Manna<br>Margaret S Robinson |

The funders had no role in study design, data collection and interpretation, or the decision to submit the work for publication.

### Author contributions

JRE, Conception and design, Acquisition of data, Analysis and interpretation of data, Drafting or revising the article; PTM, Acquisition of data, Analysis and interpretation of data, Drafting or revising the article; SN, GB, Drafting or revising the article, Contributed unpublished essential data or reagents; MSR, Conception and design, Analysis and interpretation of data, Drafting or revising the article

### Author ORCIDs

James R Edgar, http://orcid.org/0000-0001-7903-8199

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
