## [Decision Letter]

Thank you for submitting your article "Tetherin is an exosomal tether" for consideration by *eLife*. Your article has been reviewed by 2 peer reviewers, Matthew J Shurtleff (Reviewer #1); Clotilde Thery (Reviewer #2), and the evaluation has been overseen by Randy Schekman as the Senior Editor.

The reviewers have discussed the reviews with one another and the Reviewing Editor has drafted this decision to help you prepare a revised submission.

Summary:

The article by Edgar et al. 1) describes a strong increase in secretion of cholesterol- and CD63-bearing extracellular vesicles of endosomal origin, i.e. exosomes, by prevention of endosome acidification (Bafilomycin A1 treatment), and 2) demonstrates a role for endogenous tetherin to retain these EVs at the cell surface, thus preventing their complete release.

The article is a bit descriptive, with not much functional data, but is supported by beautiful and extremely informative electron microscopy analyses on the behaviour of MVB-derived EVs in HeLa cells, including correlative light-electron microcopy (CLEM), classical transmission EM, Scanning EM and cryo-immunoEM, which are very seldomly provided by most studies published in the field of EVs. Furthermore, it provides such interesting information on new tools that may be used in the future to affect exosome secretion (and thus analyse their function), that it is a very important study.

Essential revisions:

1) As mentioned in the Discussion, ILVs appear heterogenous in MVBs and thus exosomes represent a population of vesicles that likely have different cargo compositions. Given this observation, it seems relevant to look at more than a single extracellular vesicle marker to make a broad claim about exosome release. In addition to CD63 the authors should blot for other vesicle-associated proteins (e.g. Alix, TSG101, CD9, HSC70 are options, but others can be found in the literature). This is particularly important since recent work suggests that cells release multiple different vesicle sub-populations and if any of these markers are differentially affected by bafA1 treatment and/or tetherin KO, this could lead to a means of classifying vesicle sub-types.

2) And related to the above, it is becoming clear from a few recent studies that CD63 is not exclusively present on endosome-derived EVs, but it is also found in large EVs (too large to originate from MVBs) released by mast cells (Crescitelli et al., J Extracell Vesicles 2013, 2: 20677), dendritic cells but also some tumor cells (Kowal et al., PNAS 2016, 113: E968). Thus a few additional controls should be provided in Figure 2.

It is important to show that CD63 is really absent from the plasma membrane in control cells, by an EM image of control side-by-side with BafA1-treated cells in Figure 2: with such a picture, appearance of CD63+ EVs at the surface upon BafA1 treatment could not be interpreted as due to pinching off of small CD63-enriched PM domains, but instead really to MVB-derived small EVs.

3) Although the EM pictures clearly show small relatively homogenous vesicles, clearly reminiscent of the internal vesicles of MVBs, and their accumulation as clusters, also suggesting that they come from MVB fusion with the plasma membrane, this information is correlative, rather than demonstrative. Figure 2 does not convincingly demonstrate that these EVs are generated upon regurgitation of an endocytosed tracer, because in this figure the authors did not perform a real pulse-chase experiment with BSA-gold, but simply a long incubation with BSA-gold (2h): images show very little BSA-gold in MVBs and also in the surface-bound EVs, and the origin of the latter BSA-gold cannot be determined by this experiment. The authors should have incubated the cells with BSA-gold for 10-20 min, extensively washed, and then chased at 37°C for different times, as described in the original articles demonstrating MVB fusion with PM (Harding et al., J Cell Biol 1983; Raposo et al., JEM 1996).

Suggested points:

1) Does BafA1 treatment result in increased exosome secretion in multiple cell lines or only HeLa cells? This could be tested by looking at the effect of bafilomycin treatment on CD63 release into the media by other cell lines (some possibilities to test might be: HEK293T, K562, MB-MDA-231, etc.). In addition, the authors should treat the EBV-transformed B-cells shown in Figure 6 with bafilomycin and quantify exosome release by EM and/or western blotting. If exosome release is stimulated in some cells but not stimulated in others, then this would be an important finding worth reporting.

2) For the tetherin effect, do the authors suggest that only MVB-derived EVs are thus tethered to the PM, and not any other PM-derived EVs? If so, a colocalisation of tetherin with CD63 in the cells, intracellularly and at the cell surface, should be explored (Figure 3). A similar loading control of a generic EV protein, as suggested for Figure 2 above, should also be used in the WB of Figure 3.

3) In general, some WB and EM figures should, if possible, be accompanied by a quantification, or images of more cells: e.g. Figure 5 only shows one cell of each condition, how representative are these images is difficult to evaluate. How many independent times the experiments have been performed should also be indicated.

[Editors' note: further revisions were requested prior to acceptance, as described below.]

Thank you for resubmitting your work entitled "Tetherin is an exosomal tether" for further consideration at *eLife*. Your revised article has been favorably evaluated by two reviewers and Randy Schekman as the Senior editor.

The manuscript has been improved but there are some remaining issues that need to be addressed before acceptance, as outlined below. The issue raised by Reviewer #2 would appear to require another experiment or, at the least, a quantitative evaluation of data you may already have in hand. It should not be necessary to consult the referees again.

*Reviewer #1:*

The major concerns in my initial review have been satisfied in the revised manuscript.

*Reviewer #2:*

In this revised version, the authors have addressed the reviewers’ previous comments in a satisfying manner, by performing the 3 essential revisions. The MVB origin of the EVs found at the surface of BafA1-treated HeLa cells is now convincingly demonstrated.

The authors have analysed the effect of BafA1 and tetherin manipulation on secretion of other EV-associated proteins than CD63, and they show new WB in Figure 2—figure supplement 2 and Figure 3—figure supplement 3 with analysis of CD9, Tsg101 and Alix. I think that these figures must be included in the main figures, rather than a supplementary.

I am also not sure that statements like “These proteins were all somewhat affected by BafA1 treatment but less dramatically than CD63” (Results) or “unlike CD63, ALIX, Tsg101 and CD9 showed little or no increase” (Results) are acceptable in a journal with strong experimental requirements like *eLife*: quantification of the WB in control vs. Baf1-treated cells or vs. tetherin-ko cells in several independent experiments should be performed, and conclusions should not rely on a single experiment. In the blots shown Figure 2—figure supplement 2 and Figure 3—figure supplement 3, TSG101 and possibly CD9 seem to be as much upregulated by Baf1 as CD63, although the signal obtained for CD9 is surprisingly weak. Absence of upregulation by tetherin KO of these additional proteins is visible in Figure 3—figure supplement 3, but again, if only one experiment, can the authors really conclude on this potentially really interesting observation?

This request was raised in the reviewer’s comments (suggested point 3), but not really properly addressed in the answer (the authors only answer for Figure 5).

---

## [Author Response]

*Essential revisions:*

*1) As mentioned in the Discussion, ILVs appear heterogenous in MVBs and thus exosomes represent a population of vesicles that likely have different cargo compositions. Given this observation, it seems relevant to look at more than a single extracellular vesicle marker to make a broad claim about exosome release. In addition to CD63 the authors should blot for other vesicle-associated proteins (e.g. Alix, TSG101, CD9, HSC70 are options, but others can be found in the literature). This is particularly important since recent work suggests that cells release multiple different vesicle sub-populations and if any of these markers are differentially affected by bafA1 treatment and/or tetherin KO, this could lead to a means of classifying vesicle sub-types.*

The reviewer makes an excellent point, and we have now performed experiments to determine which other extracellular vesicle markers are increased following treatment with BafA1 and/or disruption of tetherin, by probing Western blots of our exosome-enriched preparations with antibodies against Alix, Tsg101, and CD9. These data are shown in Figure 2—figure supplement 2, and in Figure 3—figure supplement 3.

We find that like CD63, Tsg101, Alix, and CD9 are enriched in the 100,000x*g* pellet rather than in the 10,000x*g* pellet. In addition, for all three proteins, BafA1 increases the yield in the 100,000x*g* pellet, although the effect is not as dramatic as it is for CD63. However, none of them is substantially affected by the tetherin knockout. This suggests either that the other proteins are largely associated with non-exosomal extracellular vesicles, and/or that they are associated with a different population of exosomes from CD63. This fits in well with the reviewer’s suggestion that the markers could potentially be used to classify different vesicle subtypes, and we discuss these findings in the text in the subsections “BafA1 treatment causes an increase in exosome release”, “Tetherin links exosomes to the plasma membrane” and in the Discussion.

*2) And related to the above, it is becoming clear from a few recent studies that CD63 is not exclusively present on endosome-derived EVs, but it is also found in large EVs (too large to originate from MVBs) released by mast cells (Crescitelli et al., J Extracell Vesicles 2013, 2: 20677), dendritic cells but also some tumor cells (Kowal et al., PNAS 2016, 113: E968). Thus a few additional controls should be provided in Figure 2.*

*It is important to show that CD63 is really absent from the plasma membrane in control cells, by an EM image of control side-by-side with BafA1-treated cells in Figure 2: with such a picture, appearance of CD63+ EVs at the surface upon BafA1 treatment could not be interpreted as due to pinching off of small CD63-enriched PM domains, but instead really to MVB-derived small EVs.*

We now include an image of control cells surface-labelled for CD63 to go alongside the image of BafA1-treated cells (Figure 2). Our image shows that CD63 surface labelling in control cells is minimal. This can also be seen by immunofluorescence in Figure 3—figure supplement 1.

*3) Although the EM pictures clearly show small relatively homogenous vesicles, clearly reminiscent of the internal vesicles of MVBs, and their accumulation as clusters, also suggesting that they come from MVB fusion with the plasma membrane, this information is correlative, rather than demonstrative. Figure 2 does not convincingly demonstrate that these EVs are generated upon regurgitation of an endocytosed tracer, because in this figure the authors did not perform a real pulse-chase experiment with BSA-gold, but simply a long incubation with BSA-gold (2h): images show very little BSA-gold in MVBs and also in the surface-bound EVs, and the origin of the latter BSA-gold cannot be determined by this experiment. The authors should have incubated the cells with BSA-gold for 10-20 min, extensively washed, and then chased at 37°C for different times, as described in the original articles demonstrating MVB fusion with PM (Harding et al., J Cell Biol 1983; Raposo et al., JEM 1996).*

We have now carried out BSA-gold pulse chase experiments using the conditions recommended by the reviewer. HeLa cells were incubated with BSA-gold for 10 minutes, washed, and then incubated in fresh medium for either 30 minutes or 4 hours to chase the gold into endosomes or lysosomes respectively. The cells were then treated with 100nM BafA1 for 16 hours and analysed by conventional EM. The results are shown in Figure 2, replacing the images we included in our original manuscript. We found that extracellular vesicle-associated gold could be seen in the cells that had been chased for 30 minutes, but not in the cells that had been chased for 4 hours, consistent with the vesicles coming from MVB fusion with the plasma membrane.

*Suggested points:*

*1) Does BafA1 treatment result in increased exosome secretion in multiple cell lines or only HeLa cells? This could be tested by looking at the effect of bafilomycin treatment on CD63 release into the media by other cell lines (some possibilities to test might be: HEK293T, K562, MB-MDA-231, etc.). In addition, the authors should treat the EBV-transformed B-cells shown in Figure 6 with bafilomycin and quantify exosome release by EM and/or western blotting. If exosome release is stimulated in some cells but not stimulated in others then this would be an important finding worth reporting.*

We agree that it is important to find out whether the effect of BafA1 on exosome release is a common phenomenon, so we have now looked at several other cell lines in addition to HeLa cells. Three adherent cell lines (HeLa, HEK293ET, and MCF7) were grown to confluence, then half of the dishes were treated with 100nM BafA1, supernatants were collected, and exosome-enriched pellets were prepared and probed with anti-CD63. As the Western blot below shows, only the HeLa cells appeared to release appreciable amounts of CD63, either with or without BafA1 treatment. Similarly, when we performed conventional EM on HEK293 cells, both with and without BafA1 treatment, exosome-like vesicles were essentially undetectable.

We also examined suspension cultures of the 721.221 EBV-transformed B cell line. Cells were grown overnight at a concentration of 2 x 10^[6]^ cells/ml in a volume of 8 ml, either with or without BafA1, then culture supernatants were collected and exosome-enriched pellets were prepared. Again, Western blotting failed to detect any CD63 labelling. As Figure 6 shows, these cells do produce exosomes, so possibly there was insufficient material (721.221 cells do not grow up quickly and we had difficulty growing them up to high numbers), and/or the high expression of tetherin prevented the exosomes from being released from the plasma membrane.

Because these are all negative results, we have not included them in the revised version of the paper, but we do mention in the Materials and methods that so far we have only observed the BafA1 effect in HeLa cells (subsection “Regents”).

*2) For the tetherin effect, do the authors suggest that only MVB-derived EVs are thus tethered to the PM, and not any other PM-derived EVs? If so, a colocalisation of tetherin with CD63 in the cells, intracellularly and at the cell surface, should be explored (Figure 3). A similar loading control of a generic EV protein, as suggested for Figure 2 above, should also be used in the WB of Figure 3.*

We now show double labelling for tetherin and CD63 in both permeabilised and non-permeabilised cells, with and without BafA1 treatment, in Figure 3—figure supplement 1. There is a high level of colocalisation of CD63 and tetherin in the permeabilised cells, and a large increase in both proteins on the cell surface following BafA1 treatment, where again they colocalise. Our Western blotting results using antibodies against Tsg101, Alix and CD9 are shown in Figure 3—figure supplement 3. As discussed above, only CD63 shows a clear increase in the exosome-enriched preparation from tetherin knockout cells.

*3) In general, some WB and EM figures should, if possible, be accompanied by a quantification, or images of more cells: e.g. Figure 5 only shows one cell of each condition, how representative are these images is difficult to evaluate. How many independent times the experiments have been performed should also be indicated.*

Wherever possible, quantification of WB and EM has been performed. However, the cells shown in Figure 5 were transiently transfected, so there was a lot of variability from cell to cell. Therefore, we imaged two cells with similar amounts of gold labelling, because they should be expressing the wild-type and mutant constructs at similar levels. The images show a clear difference in the localisation of the two constructs, and also a clear difference in exosome clumping, and this is what we consistently found in the high-expressing cells.

[Editors' note: further revisions were requested prior to acceptance, as described below.]

*The manuscript has been improved but there are some remaining issues that need to be addressed before acceptance, as outlined below. The issue raised by Reviewer #2 would appear to require another experiment or, at the least, a quantitative evaluation of data you may already have in hand. It should not be necessary to consult the referees again.*

*Reviewer #2:*

*In this revised version, the authors have addressed the reviewers’ previous comments in a satisfying manner, by performing the 3 essential revisions. The MVB origin of the EVs found at the surface of BafA1-treated HeLa cells is now convincingly demonstrated.*

*The authors have analysed the effect of BafA1 and tetherin manipulation on secretion of other EV-associated proteins than CD63, and they show new WB in Figure 2—figure supplement 2 and Figure 3—figure supplement 3 with analysis of CD9, Tsg101 and Alix. I think that these figures must be included in the main figures, rather than a supplementary.*

*I am also not sure that statements like “These proteins were all somewhat affected by BafA1 treatment but less dramatically than CD63” (Results) or “unlike CD63, ALIX, Tsg101 and CD9 showed little or no increase” (Results) are acceptable in a journal with strong experimental requirements like eLife: quantification of the WB in control vs. Baf1-treated cells or vs. tetherin-ko cells in several independent experiments should be performed, and conclusions should not rely on a single experiment ! In the blots shown Figure 2—figure supplement 2nd Figure 3—figure supplement 3, TSG101 and possibly CD9 seem to be as much upregulated by Baf1 as CD63, although the signal obtained for CD9 is surprisingly weak. Absence of upregulation by tetherin KO of these additional proteins is visible in Figure 3—figure supplement 3, but again, if only one experiment, can the authors really conclude on this potentially really interesting observation?*

*This request was raised in the reviewer’s comments (suggested point 3), but not really properly addressed in the answer (the authors only answer for Figure 5).*

We have performed two additional independent experiments, harvesting extracellular vesicles under different conditions and probing Western blots with antibodies against CD63, Alix, Tsg101, and CD9. Signals from individual bands were quantified using ImageJ, and the means from all three experiments are now shown in the revised versions of Figure 2—figure supplement 2 and Figure 3—figure supplement 3. As you can see, the take-home message is that all three of the other proteins seem to show a similar trend to CD63, but generally the differences are less pronounced and the error bars are larger. We suspect that these three proteins may all be present in exosomes but not as enriched as CD63, and that they are probably also present in other types of extracellular vesicles.

However, we feel that Western blotting is not the best way to address this very important question. We have had to depend upon commercially available antibodies of variable quality: e.g., the anti-CD9 produces a very weak signal, even though CD9 is actually very abundant in HeLa cells, and all of the antibodies cross-react with other bands. In addition, even with the best of antibodies, quantification of Western blots is problematic. Proper quantification will require SILAC-based mass spectrometry, something intend to do in the future. Therefore, we would prefer to include the data in our supplementary figures, rather than to give the experiments too much weight by including them in the main figures.